# Steady-State Behavior and Endothelialization of a Silk-Based Small-Caliber Scaffold In Vivo Transplantation

**DOI:** 10.3390/polym11081303

**Published:** 2019-08-03

**Authors:** Helei Li, Yining Wang, Xiaolong Sun, Wei Tian, Jingjing Xu, Jiannan Wang

**Affiliations:** 1National Engineering Laboratory for Modern Silk, College of Textile and Clothing Engineering, Soochow University, Suzhou 215123, China; 2Division of Bioscience, University College London, London WC1E 6BT, UK

**Keywords:** silk, small caliber, artificial vascular graft, patency, endothelialization

## Abstract

A silk-based small-caliber tubular scaffold (SFTS), which is fabricated using a regenerated silk fibroin porous scaffold embedding a silk fabric core layer, has been proved to possess good cell compatibility and mechanical properties in vitro. In this study, the endothelialization ability and the steady-state blood flow of SFTSs were evaluated in vivo by implanting and replacing a common carotid artery in a rabbit. The results of the color doppler ultrasound and angiographies showed that the blood flow was circulated in the grafts without aneurysmal dilations or significant stenoses at any time point, and ran stronger and close to the autologous blood vessel from one month after implantation. The SFTSs presented an initial tridimensionality without being distorted or squashed. SEM and immunohistochemistry results showed that a clear and discontinuous endodermis appeared after one month of implantation; when implanted for three months, an endothelial layer fully covered the inner surface of SFTSs. RT-PCR results indicated that the gene expression level of CD31 in SFTSs was 45.8% and 75.3% by that of autologous blood vessels at 3 months and 12 months, respectively. The VEGF gene showed a high expression level that continued to increase after implantation.

## 1. Introduction

Vascular bypass grafting is a major mode of treatment for ischemic heart disease and peripheral vascular disease via autotransplantation, allotransplantation, and artificial vascular transplantation. Large caliber artificial vessels have been a successfully realized clinical applicationfor a long time; however, small-caliber artificial vascular transplantation has not been as successfully applied. In the clinic, small-caliber artificial vascular transplantations have thus far been performed by autotransplantation, such as heart bypass surgery using the saphenous vein or mammary artery [1,2]. However, not all patients can provide a healthy small-caliber vessel, and transplantation can also cause the vessel’s integrity to deteriorate.

Chemically synthesized polymers of expanded poly-tetrafluoroethylene or Dacron have performed very well in large caliber artificial vascular transplantation but have had poor patency and poor histocompatibility in small vascular transplantation [3,4]. The inability of such grafts to achieve early endothelialization together with the compliance mismatch between the grafts and the native vessels leads to the adherence of blood proteins and the activation of the clotting mechanisms [5,6]. In order to realize long-term patency and structural integrity in small vascular transplantation, an ideal vascular graft should have an anti-thrombotic surface and be able to rapidly form an endodermis. Owing to its excellent cellular compatibility, versatile processability in different formats, and controllable biodegradability, silk fibroin has been studied extensively for application in tissue regeneration and repair [7,8,9,10,11] and as a drug delivery carrier [12,13].

The silk fibroin molecule consists of 20 kinds of α-amino acids, in which the numbers of acidic amino acid (such as Asp and Glu) residues are greater than the numbers of alkaline amino acid (Lys) residues, leading the silk fibroin to exhibit a strong electronegativity. The inner surface of natural blood vessels has a negative charge, which protects blood vessel cells and regulates the balance of the coagulation system. Therefore, silk fibroin is more suitable for fabricating artificial vessels. In recent years, researchers have paid more attention to developing fibroin-based small-caliber artificial vessels using a variety of fabrication techniques, such as electrospinning and salt leaching. These products exhibit good biocompatibility in vivo and in vitro [14,15]. However, long-term stability remainsa major problem due to the collapse of these vascular grafts. Poor mechanical properties and intimal hyperplasia are the major reasons for vascular occlusion. In our previous work, a porous tubular scaffold prepared using silk fabric and regenerated silk fibroin (SFTS) was found to possess excellent mechanical properties [16,17,18] that helped to prevent vascular graft collapse, and could be a good substrate to support the ingrowth or co-culture of human umbilical vein endothelial cells (HUVECs), human aorta vascular smooth muscle cells (HAVSMCs), and human arterial fibroblasts (HAFs). The silk fibroin substrate can promote adhesion and proliferation of HUVECs and HAFs, while slightly inhibiting HAVSMCs proliferation; this is a pretty interesting result and may help to prevent intimal hyperplasia in artificial blood vessels [16,19,20].

In comparison, rapid endothelialization plays a decisive role in realizing the patency of small-caliber vascular grafts. The native blood vessel wall is composed of three different layers: the intima, media, and adventitia [21]. The intima has a one-cell-thick lining called an endothelium that is responsible for reducing thrombosis and controlling blood flow [22,23]. Although an electrostatically spun silk-based small-caliber vascular graft demonstrated near complete endothelialization after six weeks of implantation [24], the physical structure (such as pores) of the surfaces/interface can affect the endothelial cells’ homing ability [19]. The endothelialization capacity of materials affects the subsequent fate of tissue repair after in vivo transplantation of small-caliber vascular grafts. Any attempt to develop a new small-caliber vascular graft must be explored with endothelialization studies that use appropriate animal models. Considering the size of diseased small vessels that commonly occur in the clinic and the kinship, numbers, and cost of animals, we thought a rabbit would be a more suitable receptor for transplant research rather than the mouse model that is selected in many studies. The purpose of this study is to observe the steady-state blood flow for 12 months after the replacement of the rabbit’s common carotid artery, and explore the speed of endothelialization on the inner surface of the previously described porous silk composite scaffold.

## 2. Materials and Methods

### 2.1. Preparation of Silk Fibroin Aqueous Solution

*Bombyx mori* silk fibroin solution was prepared as described previously [25]. Briefly, 0.06% of Na_2_CO_3_ solution was used to treat raw silk to remove sericin. Degummed silk fibers were dissolved in CaCl_2_ solution. After dialyzing against distilled water at 4 °C for 3 days, the silk fibroin aqueous solution was concentrated to 8% (*w*/*w*).

### 2.2. Preparation of SFTSs

SFTSs were fabricated as described previously [26]. A silk tubular fabric was braided by twisting 24 shares of 120 denier degummed threads on a braiding machine (Shanghai Hakao, Shanghai, China). After coating silk fibroin on the tubular fabrics, the specimens were freeze-dried at −40 °C to obtain SFTSs with an inner diameter of 3.0 mm (Figure 1), which were crosslinked by polyethylene glycol diglycidyl ether (PEG-DE; MW500D; Sigma, St. Louis, MO, USA). After immersion in deionized water to remove unreacted residues, all SFTSs were sterilized by Co^60^ gamma irradiation and kept at 4 °C.

### 2.3. SFTS Implantation

Vascular grafts (15 mm in length and 3 mm in inner diameter) were implanted to replace the rabbit’s common carotid artery. The rabbits (New Zealand white rabbit, 2.5 kg, Laboratory Animal Research Center, Soochow University, Suzhou, China) were anesthetized by pentobarbital sodium (Tiandz, Beijing, China) at 30 mg/kg via intravenous injection. The animal model was approved by the ethical committee of Soochow University (No. ECSU-2019000135). A midline incision in the neck was made, and the common carotid artery was dissected. After infusion of 100 U/kg heparin (130 U/mg; Sinopharm, Beijing, China) intravenously, the common carotid artery was excised and replaced by SFTSs using two end-to-end anastomoses, and sutured with 8−0 nylon sutures (Jinhuan, Shanghai, China). The surgical wound was closed in layers with absorbable 3−0 silk sutures (Jinhuan, Shanghai, China) (Figure 1). Rabbits were monitored during recovery from anesthesia. Rabbits were followed up for 2 weeks (2 W), 1 month (1 M), 3 months (3 M), 6 months (6 M), and 12 months (12 M) (*n* = 6 per time point). The autogenous common carotid arteries of rabbits were used as controls.

### 2.4. Color Doppler Sonography and Digital Subtraction Angiography

After implantation, color doppler sonography was used to observe the dynamic blood flow in the SFTSs using the MyLabClassC doppler ultrasound equipment (Esaote, Genoa, Italy). Digital subtraction angiography was also used to evaluate the patency using an LA 523 contrast probe by injecting a SonoVue^®^ contrast agent (Bracco, Milan, Italy).

### 2.5. Scanning Electron Microscopy (SEM)

The endodermis of SFTSs taken from rabbit necks at different time points was observed under an SEM (Hitachi S-4800, Tokyo, Japan) with an accelerating voltage of 3 kV. Samples were soaked three times with phosphate-buffered saline (PBS) (pH=7.4) and dehydrated by exposure to a gradient concentration of ethanol (50, 70, 80, 90, 95, and 100% for 15 min, respectively). Finally, the samples were fixed on copper plates and sputter-coated with gold, then observed by SEM.

### 2.6. Histological Evaluation

For the histological evaluation, the grafts were harvested at the designed time point and fixed in 10% neutral buffered formalin for 2 days. Specimens were dehydrated in a series of graded ethanol baths, made transparent in a mixture of anhydrous ethanol and xylene (1:1, *v*/*v*), and then embedded in paraffin and sectioned into 5-μm-thick slices. These sections were used for further histological and immunohistochemical analysis as described below. For hematoxylin and eosin (H&E) staining: after deparaffinization by xylene and hydration by a graded concentration of ethanol (100, 95, 90, 80, and 70%), the sections were immersed in hematoxylin (Hongsheng, Shanghai, China) solution for 10 min and 1% ammonia to make them appear blue. Next, the sections were treated in a series of graded alcohol baths (70, 80, 90, and 95%) and counterstained in eosin (Sinopharm, China) solution for 2 s, then washed and immersed in 95% ethanol for coloration and were made transparent with xylene for 5 min. Finally, observation was performed with an IX51 inverted microscope (Olympus, Tokyo, Japan).

### 2.7. Immunohistochemical Assay

CD31 and VEGF were used to mark vascular endothelial cells (VECs). After deparaffinization and hydration, the sections were dipped in 3% (*v*/*v*) H_2_O_2_ to inactivate endogenous peroxidase, then heated at 9 °C in 10 mM sodium citrate (pH = 6.0) for antigen retrieval and blocked using 5% (*w*/*v*) bovine serum albumin (BSA). After removing BSA, sections were incubated overnight at 4 °C with a 1:800 diluted mouse anti-rabbit VEGF primary antibody (Abcam, Cambridge, UK) or mouse anti-rabbit CD31 primary antibody (Abcam). Biotinylated goat anti-mouse IgG (1:500 dilution) (Gene technology, Shanghai, China) was used as a secondary antibody. Color detection was performed by 3,3-N-diaminobenzidine tertrahydrochloride horseradish peroxidase color development kit (Gene technology) according to the manufacturer’s instructions. Finally, all sections were counterstained with hematoxylin and observed on an IX51 inverted microscope (Olympus). Immunofluorescence analysis was performed with 1:500 diluted Alexa Fluor^®^ 488 goat anti-mouse IgG secondary antibody (Abcam). All sections were counterstained with 4,6-diamidino-2-phenylindole (DAPI, Sigma) and observed on a FV1000 confocal laser scanning microscope (Olympus) at an excitation wavelength of 488 nm.

### 2.8. RNA Extraction and Real-Time PCR

Gene expression levels of CD31 and VEGF for VECs from regenerative endothelial tissue covered on the inner surface of implanted SFTFs were analyzed by a quantitative real-time reverse transcriptase-polymerase chain reaction (RT-PCR). According to the supplier’s instructions, total RNA was extracted from the scaffolds using a total RNA purification kit (Sangon Biotech, Shanghai, China), then reverse-transcribed using a cDNA synthesis kit (M-MuLV, Sangon Biotech). RT-PCR were performed as follows: 95 °C/15 min for pre-denaturation, 40 cycles of 95 °C/30 s for denaturation, (54–60 °C)/30 s for annealing and 72 °C/30 s for elongation, and carried out on a Step One Plus Real-Time PCR System (Applied Biosystems, Foster City, CA, USA). The primers and the annealing temperature of each gene are shown in Table 1. The housekeeping gene GAPDH was used as the reference transcript. The 2^−ΔΔCT^ method was used for the quantitative analysis.

## 3. Results and Discussion

### 3.1. Dynamic Patency and Geometrical Morphology of SFTSs

Keeping an unobstructed blood flow is the most important factor that determines whether an artificial blood vessel can be applied or not. In our work, firstly, we investigated the endurance of the dynamic patency of SFTSs during and up to 12 months following surgery. We visually observed that the appearance of the lumen surface of the removed tubular scaffolds was very clean and smooth with no signs of aneurysmal dilations or significant stenoses at any time point. The SFTSs retained their initial tridimensionality and smooth inner surface.

In the early stages of implantation, the blood flow signals became relatively weak when flowing from natural blood vessels through an exogenous material surface (Figure 2A,B). Three months after implantation, the diastolic blood flow velocity rans tronger and remained stable until 12 months after implantation, and was close to that of the autologous blood vessel (Figure 2C–F). The SFTSs retained their complete shape without being distorted or squashed, but showed a reduced inner diameter that changed from 3.0 mm to 2.5 mm after 2 weeks (Figure 2G,I). The reason for this was that the scaffolds were oppressed by the surrounding tissue while short of the autonomic diastolic function, although they were found to possess excellent mechanical properties [17,18]. The inner diameter of SFTSs reverted to 2.9 mm after 1 month of implantation, and stabilized at 3.2 mm without any significant change after 6 months of implantation (Figure 2H,I). Because, after 1 month of implantation, the porous silk fibroin was gradually displaced by a large number of pyknotic extracellular matrices, collagen fibers and elastic fibers formed and were distributed in an orderly way in SFTSs (Data not shown), leading to the inner diameters reverting and stabilizing.

The frequency spectrum of blood flow showed that the systolic and diastolic blood flow velocities became strong and stable spontaneously via the inner surface of the SFTSs, close to the autologous blood vessels on the opposite side. The results indicated that the SFTSs had a satisfactory ability against thrombus formation. Regenerated silk fibroin possesses better blood compatibility in comparison with poly-tetrafluoroethylene and Dacron [27]. Certainly, scholars have put forth more effort to improve the antithrombogenicity of small-caliber vascular transplantation by modification with heparin [28,29] or hirudin [27,30]. Here, the SFTSs with a fabric design in the medium showed remarkable antithrombotic properties, and satisfactory resistance and resilience to tissue oppression or blood pressure, owing to their excellent mechanical properties [17,18]. These results indicate that the SFTSs are feasible to use for the replacement of clinically pathological small-caliber vessels.

### 3.2. Microscopic Observation of SFTSs

The inner surface of the SFTSs removed at each time point after implantation were observed by SEM, as shown in Figure 3. At 1 month in vivo, a thin extracellular matrix layer of endothelial-like tissue fully covered the inner surface of the SFTSs. VECs occupied the inner surface and adopted a pebble-like morphology with many interconnections (Figure 3A). After 3 months of implantation, it can be seen that a completely fused endothelial layer formed on the inner surface, and linear folds appeared on the endothelium layer along the bloodstream (Figure 3B,C). Fully spread VECs were uniformly and orderly distributed in the extracellular matrix. The diastolic blood flow signals at 3 months were enhanced, owing to the endodermis that formed and the improvement in blood fluidity (Figure 2C). Rapid adhesion and proliferation of VECs are crucial to realize endothelialization after artificial vascular grafting, leading to the prevention of thrombosis and the maintenance of long-term patency [31]. The VECs in vascular grafts are derived from the differentiation by blood circulation of autologus endothelial cells or endothelial progenitor cells (EPCs) [6,7,32]. Six months after implantation into the body, a regular groove structure similar to the autologous blood vessel (Figure 3D) formed, suggesting potential vascular tissue regeneration in vivo. These results indicate that the SFTSs are favorable for endothelialization, which is not only related to the biocompatibility of the silk fibroin, but also to the inner surface structure with a few micropores (Figure 3E) [16,17,19].

### 3.3. Histological Analysis of Endothelialization

Because rapid re-endothelialization can be of vital importance, the ability of the SFTSs to endothelialize was verified by a histologic analysis and an immunohistochemical analysis. After implantation for 2 weeks, H&E staining showed that a large number of VECs had migrated to and engrafted on the inner surface of SFTSs, as shown in Figure 4A. At 1 month, the VECs had connected with each other by the extracellular matrix to form a cell monolayer (Figure 4B). After 3 months, a more continuous and smooth endothelial line can be seen, and an orderly arrangement of endothelial nucleus in the endodermis formed (Figure 4C,D). In addition, a large number of smooth muscle cells had infiltrated the SFTSs, and regenerative tissue similar to autogenous vascular tissue had formed at 6 months after implantation (Figure 4D,E). No cellsor extracellular deposition were observed in the non-implanted control SFTSs (Figure 4F).

### 3.4. Immunohistochemical Analysis of Endothelialization

VEGF is a stimulating factor of endothelial cell growth and induces endodermal formation and angiogenesis [33,34]. We evaluated VEGF+ VECs adhering to the inner surface of SFTSs at each time point by immunoenzyme and immunofluorescent (Figure 5) staining. We observed VEGF+ expression on the inner surface at 1 month (Figure 5A, indicated by a black arrow). From 3 months, the continuous locus showing VEGF+ expression became smoother and more slender (Figure 5B,C). These results are consistent with the findings by H&E staining, and are also similarly shown in the investigation by immunofluorescent staining (Figure 5a–c), as indicated by the white arrow. In the control group of the autologous blood vessel, a continuous curve distribution of VEGF+ expression was observed owing to the regular groove structure of the endodermis (Figure 5D,d). No VEGF-positive staining was observed in the non-implanted SFTSs (Figure 5E,e).

CD31 is a membrane glycoprotein and is distributed in the endodermis to closely connect VECs to each other and maintain the integrity of the endothelial monolayer’s structure [35,36]. Figure 6 shows the CD31+ expression by immunoenzyme staining. The results are consistent with the VEGF+ expression results. A large amount of CD31 protein was irregularly deposited on the inner surface of SFTSs at 1 month (Figure 6A), as indicated by the black arrow. After implantation for 3 months, the CD31+ expression levels increased with a smoother and more slender CD31+ line forming continuously (Figure 6B,C), which gradually became similar to the autologous blood vessel (Figure 6D). No CD31-positive staining was observed in the non-implanted SFTSs (Figure 6E).

The immunohistochemistry results for VEGF and CD31 expression showed that VECs preferred to rapidly adhere to the inner surface of the SFTSs, and exhibited high proliferation activity and a secretory capacity in the extracellular matrix. A complete endodermis was formed at 3 months after surgery, and it self-assembled into an autologous intima-like groove structure at 6 months. These results are consistent with the SEM observation results.

### 3.5. Gene Expression Level of CD31 and VEGF

In this study, the gene expression levels of VEGF and CD31 were investigated by RT-PCR to evaluate the intima’s formation ability. In the experiment, samples were selected from the whole implant rather than only the intima. The relative expression level of CD31 reached 45.8% at 2 weeks and 75.3% at 12 months after implantation in comparison with the autologous blood vessel (Figure 7A). The relative expression level of the VEGF gene was very high at each time point (Figure 7B). VEGF was synthesized by not only endothelial cells, but also by cells such as fibroblasts [37]. The injured blood vessel induced a higher enrichment of VECs or EPCs on the inner surface of SFTSs, and the implantation materials stimulated cells to synthesize a great deal of VEGF at an earlier stage. Furthermore, the tunica adventitia of natural blood vessels contains the system of the vasa vasorum that nourishes the external tissue of the vessel wall [38]. During the process of tissue regeneration and repair, a large number of fibroblasts migrated to and proliferated in the adventitia accompanied by the formation of the vasa vasorum, leading to the enhancement of the expression of VEGF.

## 4. Conclusions

Due to the risk and impact of vascular diseases, such as atherosclerosis and angiomas, and the limited availability of autografts, tissue engineering strategies for vascular engineering have mainly focused on the creation of small-caliber artificial vessels. However, the patency of small-caliber artificial vessels still faces great challenges. A healthy endothelial layer is the only fully blood-compatible surface, and excellent mechanical properties provide a steady blood flow that can completely avoid thrombus development. The employed SFTSs exhibited good biocompatibility and excellent mechanical properties in our previous studies. The present results indicate that VECs or EPCs rapidly proliferated onto the inner surface of SFTSs, an endodermis formed at 1 month after implantation, and the blood flow velocity was able to spontaneously recover to normal and remain stable without thrombosis, restenosis, and occlusion. In addition, the SFTSs had the biodegradability to allow for full regeneration and functional recovery, and hence extend thelife expectancy of patients.

## Figures and Tables

**Figure 1 polymers-11-01303-f001:**
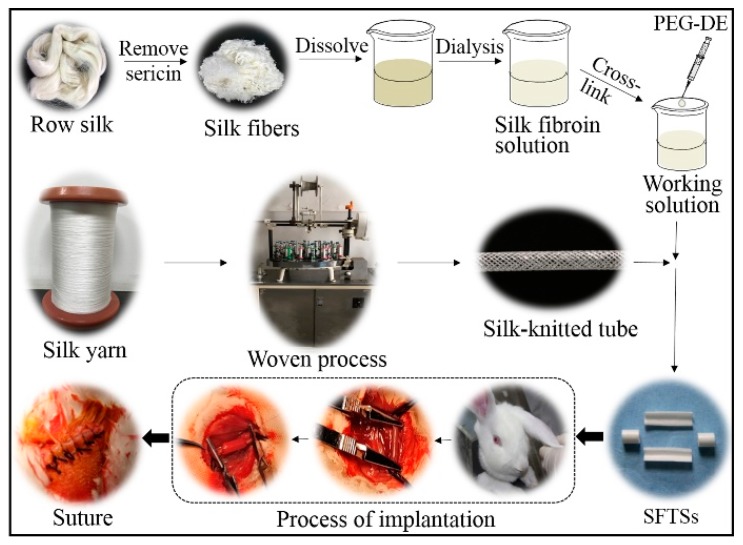
Preparation of silk-based small-caliber tubular scaffolds (SFTSs) and implantation into rabbits.

**Figure 2 polymers-11-01303-f002:**
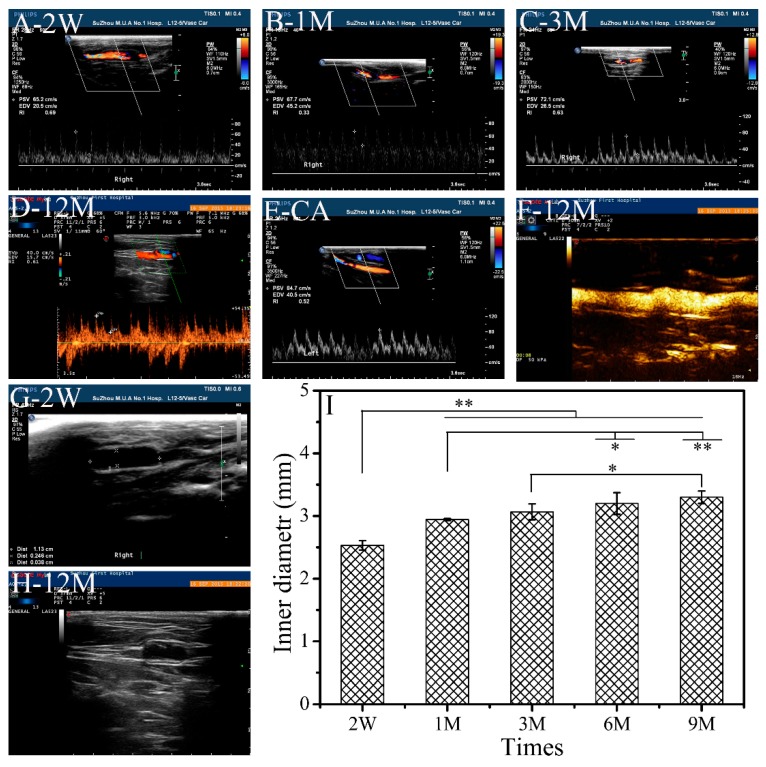
Dynamic change in blood flow (**A**–**F**) and inner diameter (**G**–**I**) of SFTSs after implantation. A–E, color doppler ultrasound for blood flow assessment; F, angiographies; G and H, color doppler ultrasound for inner diameter measurement; I, inner diameter of the SFTSs after implantation. CA, autogenous carotid artery. The data are expressed as the mean ± SD (*n* = 3). “*” means *p* < 0.05, “**” means *p* < 0.01.

**Figure 3 polymers-11-01303-f003:**
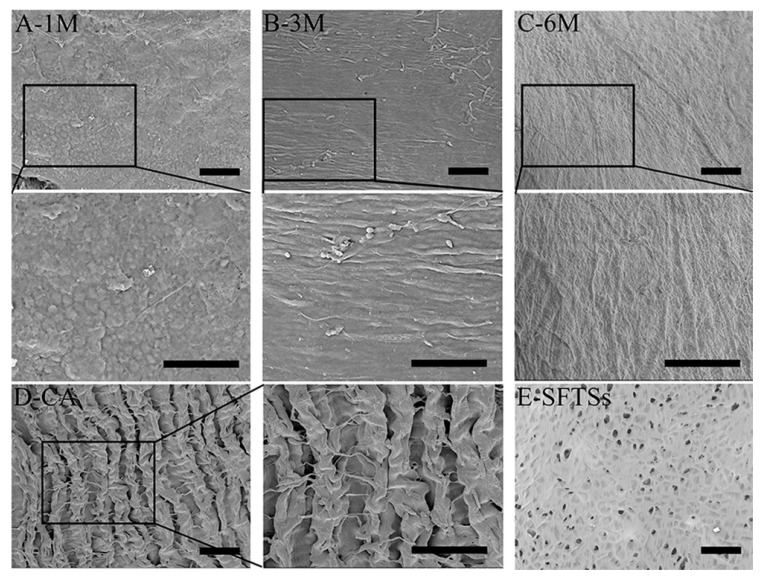
SEM images of the inner surface of SFTSs after implantation. (**A**–**C**) represent the different time points after implantation; (**D)**-CA, autogenous carotid artery; (**E**) SFTSs, non-implanted silk fibroin tubular scaffolds. The scale bar is 50 μm.

**Figure 4 polymers-11-01303-f004:**
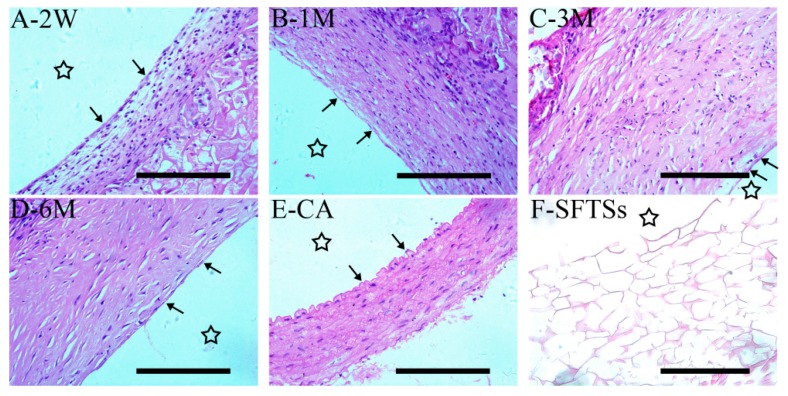
Hematoxylin and eosin (H&E) staining of an SFTS cross-section after implantation. (**A**–**D**) represent the different time points after implantation; (**E**) CA, autogenous carotid artery; (**F**) SFTSs, non-implanted silk fibroin tubular scaffolds; “→”, endothelial line. The star represents the inner cavity. The scale bar is 200 μm.

**Figure 5 polymers-11-01303-f005:**
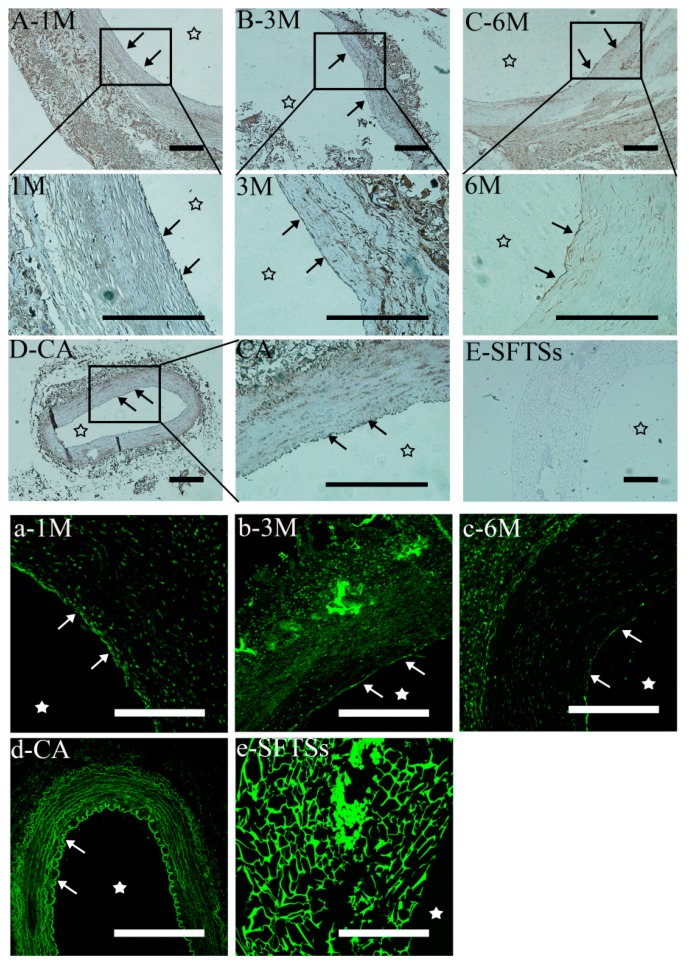
Immunohistochemistry (**A**–**E**) and immunofluorescence (**a**–**e**) analysis of endothelial cytokine VEGF+ expression. A–C and a–c represent the different time points after implantation; D-CA and d-CA, autogenouscarotid arteries; E-SFTSs and E-SFTSs, non-implanted silk fibroin tubular scaffolds; “→”, endothelial line. The star represents the inner cavity. The scale bar is 200 μm.

**Figure 6 polymers-11-01303-f006:**
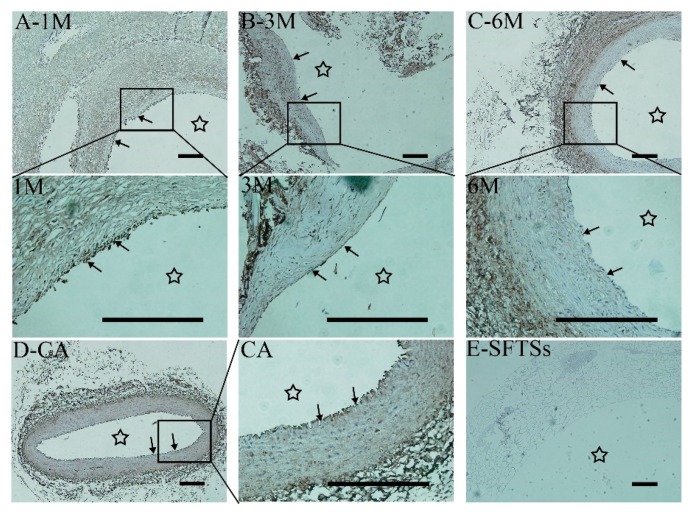
Immunohistochemistry of endothelial cytokine CD31+ expression. (**A**–**C**) represent the different time points after implantation; (**D**) CA, autogenous carotid artery; (**E**) SFTSs, non-implanted silk fibroin tubular scaffolds; “→”, endothelial line. The star represents the inner cavity. The scale bar is 200 μm.

**Figure 7 polymers-11-01303-f007:**
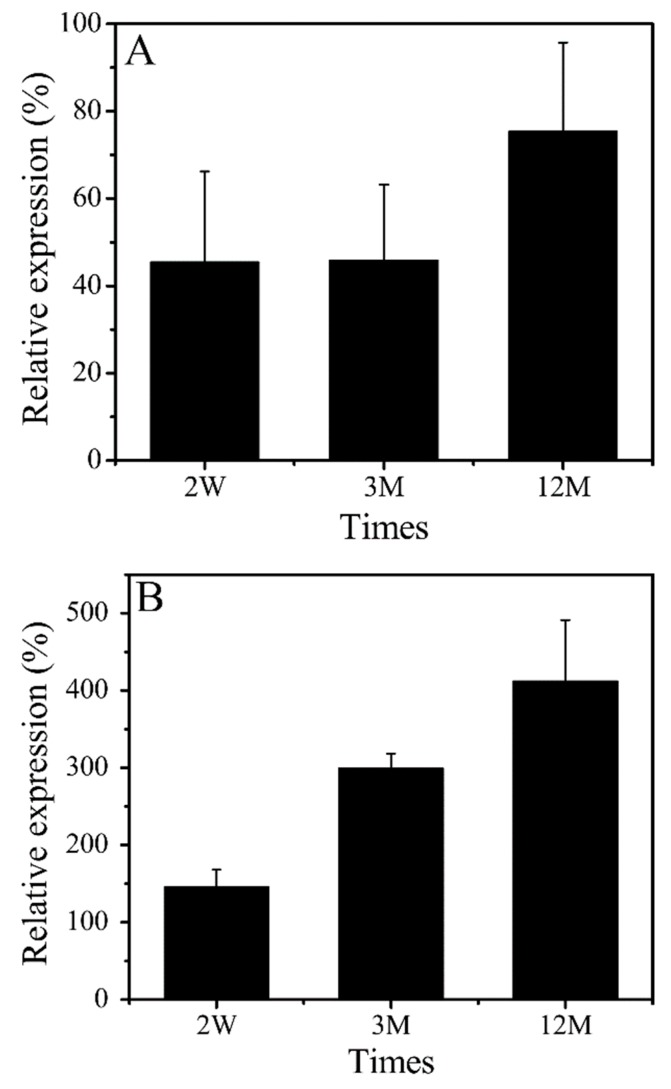
Quantitative analysis of gene expression level. (**A**), CD31; (**B**), VEGF.

**Table 1 polymers-11-01303-t001:** The primer sequence and product length of each cytokine.

Gene	Primer Sequence (5′-3′)	Product Length (bps)	Annealing Temperature (°C)	Amplification Efficiency (%)
GAPDH	GTCACTGGTGGACCTGACCT AGGGGTCTACATGGCAACTG	420	60	105.677
VEGF	GCTCAGAGCGGAGAAAGCAT GCAACGCGAGTCTGTGTTTT	80	54	108.152
CD31	GGTGGATGAGGTCCAGATTTC CAGCACAATGTCCTCTCCAG	67	56	108.635

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
