# Peer review of "Steady-State Behavior and Endothelialization of a Silk-Based Small-Caliber Scaffold In Vivo Transplantation"

_polymers, 2019, doi:10.3390/polym11081303_

Round 1

Reviewer 1 Report

This manuscript show Steady-State Behavior and Endothelialization of Silk Small-Caliber Scaffold in Vivo Transplantation. Good results were presented, but lack physicochemical characterization to confirm if the Preparation of Silk Fibroin Aqueous Solution and SFTs . Degradation tests would be very good also to verify the in vitro behavior in the simulated fluids. Authors should add on methodology information about ethics committee. No references are used in the conclusions, these have to be based on the results presented in the article. And the references must be formatted according to the norms of the journal.

Author Response

1. This manuscript show Steady-State Behavior and Endothelialization of Silk Small-Caliber Scaffold in Vivo Transplantation. Good results were presented, but lack physicochemical characterization to confirm if the Preparation of Silk Fibroin Aqueous Solution and SFTS.

Answer: The physicochemical characterization of silk fibroin aqueous and SFTS has been studied in our previous studies (such as reference Nos 16, 17, 18).

2. Degradation tests would be very good also to verify the in vitro behavior in the simulated fluids.

Answer: The degradation of porous materials or fibers has been reported. The SFTSs has achieved long-term patency and in situ tissue regeneration in vivo. The matching between scaffold degradation rate and tissue regeneration rate in vivo is very important. We are focusing on studying and regulating the matching between the scaffold degradation and tissue regeneration in the next works. 

3. Authors should add on methodology information about ethics committee.

Answer: We have revised.

4. No references are used in the conclusions, these have to be based on the results presented in the article.

Answer: We have revised in the conclusions.

4. The references must be formatted according to the norms of the journal.

Answer: We have revised according to the norms of the journal.

Reviewer 2 Report

The aim of this study was to fabricate a silk-based small caliber tubular scaffold and be used to replace the common carotid artery in the rabbit. Then the endothelialization ability, the blood flow, tissue growth and gene expression were evaluated. Their conclusion suggested that the artificial vessel not only provide steady mechanical property for blood flow but also it provide an environment that cells can rapidly growth on the inner surface. Overall, the experiments were well done and it is a scientifically sound piece of work However, the reviewer has the following suggestions and comments before it can be published:  

1.     For the animal experiments, please provide IACUC number. In addition, how many animals were used in this study (or in each observation time)? From Fig. 2I, the data expressed as mean and SD. How many samples were collected to calculate these mean values? Since mean and SD can be calculated, please perform statistic analysis on these data and discuss the differences in these values.

2.     Fig. 3D and E were not mentioned in the text and figure legend. Please provide a more detail description for “E”. The same comments also for Fig. 4F, Fig. 5E/e and Fig. 6E. In the text, I cannot find any description or explanation regarding these sub-figures. To Please add them in the text

3.     On line 230, “observed in the negative control of SFTSs”. Please make sure “SFT” was used as a negative control group, or it is only negative to express VEGF.

4. Why the author choice to test the expression of CD31 and VEGF. There is no explanation on the motivation for these tests. Please add the background of why the author tested these two genes. Please also added, a least, a paragraph to discuss the findings of CD31 in your experiments. 

Author Response

1. For the animal experiments, please provide IACUC number. In addition, how many animals were used in this study (or in each observation time)? From Fig. 2I, the data expressed as mean and SD. How many samples were collected to calculate these mean values? Since mean and SD can be calculated, please perform statistic analysis on these data and discuss the differences in these values.

Answer: (1) The ethics consent of animal has been sent to the editor.

(2) We have added the number of animals in the section 2.3.

(3) We have been added the number of samples in the Fig. 2 and supplemented the explanation.

2. Fig. 3D and E were not mentioned in the text and figure legend. Please provide a more detail description for “E”. The same comments also for Fig. 4F, Fig. 5E/e and Fig. 6E. In the text, I cannot find any description or explanation regarding these sub-figures. To Please add them in the text.

Answer: We have revised.

3. On line 230, “observed in the negative control of SFTSs”. Please make sure “SFTSs” was used as a negative control group, or it is only negative to express VEGF.

Answer: We have revised.

4. Why the author choice to test the expression of CD31 and VEGF. There is no explanation on the motivation for these tests. Please add the background of why the author tested these two genes. Please also added, a least, a paragraph to discuss the findings of CD31 in your experiments. 

Answer: We have revised. Besides, CD31 and VEGF are the most useful molecular markers of VECs.

Round 2

Reviewer 2 Report

The author answered most questions that I concerned. However, there is some minor modified is needed.

The ethical document number or committee agreement number that proof the animal study performed was following the 3R policy of Declaration of Helsinki should be clearly listed in the Material and Method of the manuscript, but not "set to the Editor".  

In my second comments, I mentioned that Figures 3D, 3E, 4F, 6E were not cited in the text. Also, there is no description of these figures. The authors stated that they have revised these statements. However, I can not find the revision in the revised manuscript. Please clearly identify how and where the author revised.  

Author Response

The ethical document number or committee agreement number that proof the animal study performed was following the 3R policy of Declaration of Helsinki should be clearly listed in the Material and Method of the manuscript, but not "set to the Editor".  

Answer: We have added the ethical document number labelled with yellow highlighting in the section 2.3.

In my second comments, I mentioned that Figures 3D, 3E, 4F, 6E were not cited in the text. Also, there is no description of these figures. The authors stated that they have revised these statements. However, I can not find the revision in the revised manuscript. Please clearly identify how and where the author revised.

Answer: We have cited and explained the Figures 3D, 3E, 4F and 6E in the text, and added yellow highlightings in the sections 3.2, 3.3 and 3.4, respectively.
